# Naringin’s Alleviation of the Inflammatory Response Caused by *Actinobacillus pleuropneumoniae* by Downregulating the NF-κB/NLRP3 Signalling Pathway

**DOI:** 10.3390/ijms25021027

**Published:** 2024-01-14

**Authors:** Qilin Huang, Wei Li, Xiaohan Jing, Chen Liu, Saad Ahmad, Lina Huang, Guanyu Zhao, Zhaorong Li, Zhengying Qiu, Ruihua Xin

**Affiliations:** 1Lanzhou Institute of Husbandry and Pharmaceutical Sciences of Chinese Academy of Agricultural Sciences (CAAS), Lanzhou 730050, China; 82101215691@caas.cn (Q.H.); xhanjing306@gmail.com (X.J.); 82101225773@caas.cn (C.L.); saadahmad.uaf@gmail.com (S.A.); zhaogy517@gmail.com (G.Z.); lizr@st.gsau.edu.cn (Z.L.); 2Engineering and Technology Research Center of Traditional Chinese Veterinary Medicine of Gansu Province, Lanzhou 730050, China; 3Key Laboratory of Veterinary Pharmaceutical Development of Ministry of Agriculture and Rural Affairs of China, Lanzhou 730050, China; 4Lanzhou Center for Disease Control and Prevention, Lanzhou 730050, China; liwei@lzcdc.cn; 5State Key Laboratory of Applied Organic Chemistry, School of Pharmacy, Lanzhou University, Lanzhou 730013, China; 220220943870@lzu.edu.cn

**Keywords:** *Actinobacillus pleuropneumoniae* (APP), inflammatory injury, NLRP3 inflammasome, protein interactions, naringin (NAR), anti-inflammatory mechanism

## Abstract

*Actinobacillus pleuropneumoniae* (APP) is responsible for causing Porcine pleuropneumonia (PCP) in pigs. However, using vaccines and antibiotics to prevent and control this disease has become more difficult due to increased bacterial resistance and weak cross-immunity between different APP types. Naringin (NAR), a dihydroflavonoid found in citrus fruit peels, has been recognized as having significant therapeutic effects on inflammatory diseases of the respiratory system. In this study, we investigated the effects of NAR on the inflammatory response caused by APP through both in vivo and in vitro models. The results showed that NAR reduced the number of neutrophils (NEs) in the bronchoalveolar lavage fluid (BALF), and decreased lung injury and the expression of proteins related to the NLRP3 inflammasome after exposure to APP. In addition, NAR inhibited the nuclear translocation of nuclear factor kappa-B (NF-κB) P65 in porcine alveolar macrophage (PAMs), reduced protein expression of NLRP3 and Caspase-1, and reduced the secretion of pro-inflammatory cytokines induced by APP. Furthermore, NAR prevented the assembly of the NLRP3 inflammasome complex by reducing protein interaction between NLRP3, Caspase-1, and ASC. NAR also inhibited the potassium (K^+^) efflux induced by APP. Overall, these findings suggest that NAR can effectively reduce the lung inflammation caused by APP by inhibiting the over-activated NF-κB/NLRP3 signalling pathway, providing a basis for further exploration of NAR as a potential natural product for preventing and treating APP.

## 1. Introduction

*Actinobacillus pleuropneumoniae* (APP) is responsible for causing porcine pleuropneumonia (PCP), which is characterized by acute haemorrhagic necrotizing pneumonia, as well as chronic fibrinous pleuropneumonia [1]. The spread of APP between pig herds occurs mainly through direct contact, aerosols, and manure [2]. The various virulence factors of APP trigger cascade reactions after infection, leading to an “inflammatory storm” characterized by lung haemorrhage, necrosis, and fibrinous exudate [1], resulting in a mortality rate of up to 80–100%. Commonly used antibiotics like florfenicol, tetracycline, and enrofloxacin are usually employed for treating PCP. However, the effectiveness of antibiotics in preventing and controlling the disease is being diminished by growing bacterial resistance [3]. Additionally, the frequent international trade of newly developed species has led to the emergence of more complex APP serotypes, resulting in poor cross-immunity between different serotypes, which has created a bottleneck in vaccine prevention and control [4,5]. As a result, the development of natural drugs for preventing and controlling APP has become a topic of increasing interest among researchers.

Naringin (NAR) is a dihydroflavonoid compound abundantly found in the dried unripe peel of the citrus “Tomentosa” or “*mandarin* (L.)” [6,7], which has traditional pharmacological traceability and is known for its antitussive and expectorant effects [8]. Recent studies have revealed that NAR has beneficial consequences including anti-inflammatory, antioxidant, and anti-apoptotic responses [9,10,11]. After NAR enters the body, its active metabolite naringenin protects endothelial cells from apoptosis and inflammation by regulating the Hippo-YAP pathway [12]. Most notably, NAR has shown effectiveness in repairing inflammatory damage in the respiratory tract, and reducing sputum secretion and inflammatory infiltration in lungs [13]. NAR can improve lung inflammation by inhibiting the release of interleukin (IL)-8, leukotriene B4, tumour necrosis factor (TNF)-α, and persistent neutrophilic infiltration due to infection [14]. Furthermore, NAR can prevent ovalbumin-induced airway inflammation by regulating the T1/T2 cell ratio [15]. However, it has not been confirmed whether NAR has the ability to repair the inflammatory damage caused by APP.

Porcine alveolar macrophages (PAM) are widely distributed on the alveolar and bronchial surfaces, forming a vital defence barrier for innate lung immunity [16]. When APP invades, pathogen-associated molecular patterns (PAMP) and damage-associated molecular patterns (DAMP) in PAM are activated, causing it to release an excess of inflammatory cytokines [13,16]. When lung tissue is overloaded with inflammatory factors, it activates the NLRP3 inflammatory vesicle classical/non-classical pathway in turn, which mediates apoptosis as well as cellular pyroptosis via the caspase-1, caspase-8, and caspase-9 pathways, further exacerbating inflammation [17,18]. A large number of studies have demonstrated that NLRP3 inflammasomes are associated with the pathogenesis of a variety of respiratory diseases [19], such as rhinitis [20], asthma [21], and chronic obstructive pulmonary disease (COPD) [22]. It has been found that the virulence factors Lipopolysaccharide (LPS) [23] and *Actinobacillus pleuropneumoniae* toxin (Apx) [17] of APP could initiate activation of the NLRP3 inflammasome signalling pathway, which is closely associated with the lung inflammatory injury of PCP. Therefore, in the present study, we investigated the role of NAR in repairing inflammatory lung injury through an APP-induced mice pneumonia model and a PAM inflammatory cell model, and elucidated the mechanism by which NAR exerts anti-inflammatory effects by inhibiting the NF-κB/NLRP3 inflammasome signalling pathway.

## 2. Results

### 2.1. Effects of NAR on Body Weight, Food Intake, and Macroscopic Lesions of Lung Tissue after APP Infection

In the study, we observed weight changes and symptoms in mice after APP infection. Before infection, all groups of mice showed continuous weight gain from day 1 to day 7; after APP infection on the 7th day, mice in the model group experienced significant weight loss, decreased appetite, depression, and loose stool. The results showed that the intervention of NAR was able to reverse the weight loss (Figure 1A). Before APP infection, there were no differences in the food intake among the groups of mice. However, the model group showed a decreasing trend in food intake compared to the control group after infection, and NAR administration reversed the feed intake decline to some extent (Figure 1B). The lung tissue was congested, haemorrhagic, purplish-red, and hardened in texture after APP infection and the lesions were alleviated after NAR intervention, with the highest concentration of NAR showing the most pronounced effect (Figure 1C). We further analysed total cell counts, as well as the percentage of neutrophils (NEs) and CRP in the bronchoalveolar lavage fluid (BALF). The results showed that both entire cells and NEs were significantly increased by the induced APP infection. However, after NAR intervention, both the total cell count and the percentage of NEs decreased considerably, which positively correlated with the NAR dosage (Figure 1D–F). Similarly, the levels of C-reactive protein (CRP) were significantly increased by the APP infection, and NAR intervention reduced the secretion of CRP in a dose-dependent manner (Figure 1G).

### 2.2. Effect of NAR on Histopathological Damage in the Lungs of Mice after APP Infection

After performing H&E staining on the lung tissues, we observed that the bronchial mucosa in the control group remained intact, the lung lobules had a clear structure, and the alveolar lumen was also unharmed. However, in the model group infected with APP, the lung tissues showed degeneration and necrotic shedding of alveolar epithelial cells, interstitial oedema, and haemorrhage. We also observed inflammatory cell infiltration, alveolar dilation, and interstitial widening, with a predominant infiltration of NEs. After administering NAR in different doses, we observed a reduction of tissue lesions in different treated groups. Particularly in the high-dose group (Figure 2A), the infiltration of NEs decreased, the degree of alveolar haemorrhage lessened, and the structure of the alveolar wall was restored with the increased number of inflated alveolar lumens. To judge more objectively, we used ImageJ software (Image Pro Plus version 6.0) to analyse the degree of inflammatory cell infiltration, lung parenchyma, and alveolar wall thickness in the mice’s lung tissue. The results showed that the model group had significantly higher levels of inflammatory cell infiltration, parenchymatous lung tissue, and alveolar thickening compared to the control group. However, after the intervention of NAR, these judging indicators were significantly reduced to a certain extent (Figure 2B). Since myeloperoxidase (MPO) is mainly found in myeloid cells—such as NEs and monocytes—and closely associated with inflammation, the results showed that MPO content was elevated in the model group and significantly reduced after treatment of NAR in a dose-dependent manner, which was consistent with the results for NEs in BALF (Figure 2C).

### 2.3. NAR Exerts Anti-Inflammatory Effects by Inhibiting the NLRP3 Inflammasome Signalling Pathway in the Lungs

NLRP3 inflammasomes are considered to be closely associated with lung inflammation. After an NLRP3 inflammasome is activated, the matured Caspase-1 lead cleaves GasderminD (GSDMD), leading to cellular pyroptosis. We examined the expression of essential proteins involved in the NLRP3 inflammasome signalling pathway using Western blotting. The results showed that the phosphorylation level of the P65 protein increased after APP infection, and the expression of NLRP3, Caspase-1, GSDMD, and IL-18 proteins was also significantly elevated (*p* < 0.05). However, administration of NAR effectively inhibited the expression of p-P65, NLRP3, Caspase-1, GSDMD, and IL-18 proteins (Figure 3A). Gene expression analysis also confirmed that NAR could reduce the gene expression of NLRP3, Caspase-1, and GSDMD (Figure 3B). Furthermore, ELISA assay results demonstrated that the protein expression of IL-1β and IL-18 significantly increased after APP stimulation, but decreased after NAR administration (Figure 3C). These findings suggest that NAR provides anti-inflammatory effects by preventing the overactivation of crucial proteins in the NLRP3 inflammasome signalling pathway induced by APP in vivo.

### 2.4. NAR Inhibits the First Signalling of NLRP3 Inflammasome Activation in PAMs

It is reported that activating the NF-κB signalling pathway is necessary to activate the NLRP3 inflammasome. Specifically, the nuclear translocation of NF-κBP65 plays a crucial role in regulating gene transcription and promoting the release of inflammatory factors such as TNF-α and IL-6 (Figure 4A). In this study, we observed that the expression levels of these inflammatory factors (TNF-α and IL-6) were significantly (*p* < 0.05) higher in PAMs caused by APP, but decreased after NAR intervention. The results were consistent at both the genetic and protein levels (Figure 4A,B). Furthermore, we analysed the protein levels of P65 in the nucleus and cytoplasm, respectively, and found that the levels of P65 in the nucleus were significantly increased after APP infection, but decreased after NAR intervention. Meanwhile, the protein levels of P65 in the cytoplasm were significantly decreased after APP infection and rebounded after NAR intervention. This result suggests that NAR could inhibit the entry of P65 into the nucleus and maintain normal levels of P65 in the cytoplasm (Figure 4C). Immunofluorescence results also supported that the expression of P65 decreased in the cytoplasm and increased in the nucleus after APP infection, which suggests that the NF-κB signalling pathway is over-activated. After NAR treatment, the amount of P65 in the nucleus was significantly reduced (*p* < 0.05) (Figure 4D). Overall, our results indicate that NAR exhibits anti-inflammatory effects by inhibiting the entry of P65 into the nucleus, reducing the activation of the first signal of NLRP3 inflammasomes.

### 2.5. NAR Exerts Anti-Inflammatory Effects by Inhibiting the NLRP3 Inflammasome Signalling Pathway in PAMs

We observed a significant increase in the expression of NLRP3 and Caspase-1 in PAMs after APP infection. However, after NAR introduction, a notable reduction in the expression of NLRP3 and Caspase-1 was observed (Figure 5A,B). Meanwhile, NAR was also found to significantly reduce the gene expression of inflammatory cytokines IL-1β and IL-18, which are closely related to the NLRP3 inflammasome. We also examined the key factor GSDMD, which is closely related to cellular pyroptosis, and whose cleavage at the Asp-275 site triggers cellular pyroptosis. The results indicate that the protein and gene expression levels of GSDMD were higher after APP stimulation compared to the control group, and the protein expression of FL-GSDMD (275) also significantly increased (*p* < 0.05). However, the expression of these factors was inhibited considerably after NAR treatment (Figure 5B). Additionally, the protein expression levels of IL-18 and IL-1β in the cell supernatant increased significantly during infection, but decreased after NAR intervention in a dose-dependent manner (Figure 5A–C). These results indicate that NAR effectively inhibits the expression of essential proteins and genes in the NLRP3 signalling pathway induced by APP infection, reduces the cleavage of GSDMD by Caspase-1, decreases cellular pyroptosis, and exerts anti-inflammatory effects.

### 2.6. NAR Exerts Anti-Inflammatory Effects by Interfering with the Assembly of NLRP3 Inflammasome Complex Proteins

In this study, we investigated the impact of NAR on the activation of the NLRP3 inflammasome’s initiating factors. Since ASC plays a crucial role in activating NLRP3 inflammasome complex proteins, we probed the level of oligomerization of the ASC protein. Our findings revealed that after APP stimulation, the level of ASC aggregation increased in PAMs compared to the control group, and the oligomerization level of ASC was significantly reduced (*p* < 0.05) to some extent after NAR intervention (Figure 6A). To further understand the assembly of NLRP3 inflammasome complex proteins, we used the immunoprecipitation technique (Co-IP) to explore the interactions between receptor protein NLRP3, effector protein Caspase-1, and junction protein ASC. The results showed that APP stimulation increased the amount that NLRP3 and Caspase-1 proteins interacted with ASC, indicating an improved assembly of NLRP3 inflammasome complex proteins. However, NAR intervention down-regulated the protein expression of NLRP3, as well as Caspase-1’s interaction with ASC, suggesting that NAR inhibited the assembly of NLRP3 inflammasome complex proteins (Figure 6B). Additionally, intracellular potassium (K^+^) efflux also triggers the activation of the NLRP3 inflammasome’s second signalling pathway. Through turbidimetric assay, we measured the intracellular K^+^ content, and the results showed that the model group had significantly reduced K^+^ content compared to the control group (*p* < 0.05). However, the cells treated with NAR showed a dose-dependent increase in intracellular K^+^ content, indicating that NAR could significantly inhibit APP-induced intracellular K^+^ efflux and subsequently inhibit the activation of the NLRP3 inflammasome (Figure 6C).

## 3. Discussion

Frequent international trade introductions have resulted in the continued spread of APP worldwide and the severe blow it has inflicted on the pig economy. The effectiveness of antibiotic and vaccine prophylaxis is greatly hampered by antibiotic resistance and poor cross-immunity between serotypes [23]. In recent years, many natural drugs have been found to possess anti-inflammatory properties. Therefore, we hope that the addition of natural plant drugs to diets will reduce the inflammatory damage caused by pathogens to the tissues of the organism, thus reducing the use of antibiotics and indirectly lowering the risk of bacterial resistance, which is ultimately conducive to healthy farming and food safety. NAR is an active flavonoid that is not only found in the peel of citrus fruits, but also widely found in grapefruit, tomatoes, cherries, and other plants. It has the advantages of being easy to extract, widely available, inexpensive, and safe to use [23]. Studies have proved that NAR has a good repairing effect on respiratory inflammatory injury, which can significantly reduce sputum secretion and inflammatory infiltration in the lungs, and also decrease goblet cell proliferation and mucus secretion [24]. Therefore, we investigated the protective effect of NAR on APP-induced lung inflammatory injury, which provides a theoretical basis for developing natural drugs for treating PCP. The results of this study showed that NAR was able to improve APP-induced symptoms such as loss of weight and appetite, alleviate lung tissue lesions in mice, and significantly reduce inflammatory cell infiltration, lung parenchymatous, and alveolar thickening in lung tissues. Through further analysis, we discovered that the mechanisms of these actions may inhibit the over-activation of the NF-κB/NLRP3 inflammasome signalling pathway, thereby exerting its effect on alleviating the inflammatory injury of the lungs.

The pathogenesis of APP is mainly through the secretion of multiple virulence factors that enter the lungs from the respiratory tract [20,25], adhere to and deposit in the alveolar epithelium, and persistently stimulate PAMs, which form an essential immune barrier in the lungs’ cell surface. Intracellular pattern recognition receptors (PRRs) activate pathogen-associated molecular patterns (PAMPs) by recognizing virulence factors, or activate damage-associated molecular patterns (DAMPs), and phosphorylate Toll-like receptors (TLRs) through the TLR4 signalling pathway; this subsequently activates the NF-κB/MAKE/AP-1 pathway, promotes the transcription of NLRP3, proIL-1β, and proIL-18, and regulates the intensity of the natural immune response [26]. When intense stimuli and inflammatory responses persist, the dynamic immune balance is disrupted. The early production of pro-inflammatory factors further promotes the release of cytokines, such as macrophage inflammatory protein 1α (MIP-1α), macrophage inflammatory protein 1β (MIP-1β), IL-6, IL-8, IL-12, etc., leading to a series of cytokine cascade responses that promote NEs to accumulate in large numbers in the alveoli, causing an inflammatory storm [18], and at the same time inducing platelets to agglutinate in the lung tissues, causing capillary rupture, leading to cell necrosis [27] and tissue damage [28]. We found that NAR significantly reduced the number of NEs in BALF of APP mice, inhibited the secretion of pro-inflammatory factors TNF-α and IL-6, reduced APP-induced lung congestion and necrosis, and reduced inflammatory cell infiltration, lung parenchyma, and alveolar thickening, which significantly ameliorated the inflammatory injury of the lungs induced by APP.

The activation of the classical NLRP3 inflammasome requires two crucial signals: signal 1 is the transcription of inflammatory cytokines, which involves the activation of the NF-κB signalling pathway, thus activating the transcription of inflammatory cytokines such as NLRP3, pro-Caspase-1, and pro-IL-1β [29], while signal 2 is the binding of NLRP3 and Caspase-1 to ASC to form NLRP3 inflammasome [30]. The LRR structural domain of NLRP3 is usually in a state of autoinhibition; its protein spatial structure will change after stimulation, and the NACHT structural domain of NLRP3 will mediate its own oligomerization, recruiting ASC through the PYD structural domain and then recruiting pro-Caspase-1 to form an inflammasome through the CARD structural domain of ASC proteins, and the activated Caspase-1 not only cleaves and activates pro-IL-1β and pro-IL-18 precursor proteins into mature forms of IL-1β and IL-18, but also cleaves Gasdermin D, which plays a vital role in cellular pyroptosis [19] in our study. We found that NAR reduced the nuclear translocation of NF-κB P65 and IL-6, as well as the TNF-α expression induced by APP, thereby inhibiting NLRP3 inflammasome’s first signalling activation. Immediately after that, we explored the effect of NAR on the second signalling of NLRP3. As oligomerization of ASC is a critical step in the activation of the second signalling of NLRP3 inflammasome, the assay results revealed that NAR was able to reduce the level of ASC oligomerization dose-dependently, and also inhibit the protein interactions between NLRP3, Caspase-1, and ASC, thus inhibiting the NLRP3 inflammasome complex protein assembly. Meanwhile, NAR reduced the expression of IL-18, IL-1β, NLRP3, Caspase-1, and GSDMD at the gene level, as well as NLRP3, Caspase-1, IL-18, GSDMD, and FL-GSDMD (275) at the protein level, suggesting that NAR could play an essential role in inhibiting the NF-κB/NLRP3 signalling pathway’s activation and also inhibit the NLRP3 inflammasome complex protein, thus exerting an inhibitory effect on the inflammatory response. Since intracellular K^+^ efflux could enhance the assembly of NLRP3 proteins and activate the NLRP3 inflammasome, the results suggest that NAR could reduce APP-induced K^+^ efflux and avoid exacerbating inflammatory responses.

In summary, NAR was able to alleviate pathological damage in mouse lung tissue due to APP-induced injury; reduce the content of NEs in BALF; and decrease the expression of crucial proteins and genes in the NLRP3 inflammasome signalling pathway in lung tissue. Meanwhile, in the APP-induced PAMs inflammatory cell model, NAR was able to not only reduce the nuclear translocation of the NF-κB P65 protein and thereby inhibit the NLRP3 first signal activation, but also inhibit the oligomerization of ASC protein, as well as reduce the protein interactions between NLRP3, Caspase-1, and ASC, thereby interfering with the assembly of the NLRP3 inflammasome complex protein. Further, NAR treatment could lessen the intracellular K^+^ efflux from the PAMs, thus exerting anti-inflammatory effects. It was shown that NAR ameliorates inflammatory injury in the lungs caused by APP induction by inhibiting the NF-κB/NLRP3 signalling pathway (Figure 7).

## 4. Materials and Methods

### 4.1. Chemicals and Reagents

NAR was purchased from Sigma Aldrich Chemical (St. Louis, MO, USA. HPLC ≥ 98%, chemical formula C_27_H_32_O_14_, molecular weight 580.54, CAS No. 10236-47-2); 1640 medium (Gibco, Grand Island, NY, USA) and trypsin were purchased from JS Biosciences Co., Ltd. (Lanzhou, Gansu Province, China); foetal bovine serum (FBS) and penicillin–streptomycin antibiotics were purchased from Thermo Fisher Scientific (Waltham, MA, USA); nuclear protein extraction kit was purchased from Solarbio Science & Technology Co., Ltd. (Beijing, China); ELISA kits for IL-6, TNF-α, IL-1β, and IL-18 were purchased from Jianglai Industry Co., Ltd. (Beijing, China); RNA extraction kit, TRIzol^®^ Reagent RT-PCR kit, and SYBR^®^ green PCR master mix were purchased from TaKaRa (Tokyo, Japan); NLRP3, Caspase-1, pro-Caspase-1, ASC, NF-κB P65, p-P65, IL-18, GAPDH, Lamin B, HRP-conjugated goat anti-rabbit IgG, HRP-conjugated goat anti-mouse primary IgG, and Alexa Flour 488-labelled anti-mouse IgG were purchased from Cell Signaling Technology, Inc. (Beverly, MA, USA); bovine serum albumin (BSA) and ECL assay kits were obtained from Thermo Fisher Scientific (Waltham, MA, USA); bovine serum albumin (BSA) and ECL assay kits were provided by Thermo Fisher Scientific (Waltham, MA, USA); and McConkay’s medium was purchased from Sigma–Aldrich (Merk, Darmstadt, Germany).

### 4.2. Bacteria, Cellular Inflammation Models, and NAR Treatment

*Actinobacillus pleuropneumoniae* (APP) isolated from swine farms and verified through Blast sequencing was conserved in our laboratory and resuscitated in TSA medium (additionally supplemented with 5% foetal bovine serum and 1% NAD) at 37 °C. PAMs were purchased from Procell Life Science & Technology Co., Ltd. (Wuhan, China), then cultured in 1640 medium, containing 1% penicillin–streptomycin and 10% FBS, at 37 °C in a humidified atmosphere containing 5% CO_2_. After the cell fusion reached about 70–80%, they were digested with trypsin and passaged for culture. The cells were pretreated with NAR (10 µg/mL, 20 µg/mL, 30 µg/mL, and 40 µg/mL) for 12 h, then stimulated with APP bacterial solution (1 × 10^7^ CFU/mL) for 1 h. After removing the bacterial solution, the cells were cultured as usual for 12 h and collected for Western blotting and RT-PCR assays, and the supernatant was collected for ELISA tests.

### 4.3. Establishment of Animal Model and Treatment with NAR

Sixty Kunming mice (18–22 g) were provided by the Lanzhou Veterinary Institute of CAAS (License No. SCXK Gansu 2023–016). Animal welfare statement: institutional ethical and animal care guidelines were observed, and all experimental processes were conducted according to the China Guide for the Care and Use of Laboratory Animals (protocol number: IACUC-157031). The mice were randomly divided into five groups, namely the control group (gavage with water), the model group (gavage with water + 10^8^ CFU/mL APP infection), the naringin low-dose group (gavage with 20 mg/kg NAR + 10^8^ CFU/mL APP infection), the naringin medium-dose group (gavage with 40 mg/kg + 10^8^ CFU/mL APP infection), and the naringin high-dose group (gavage with 80 mg/kg + 10^8^ CFU/mL APP infection) (n = 12). The drugs were administered by gavage continuously for eight d. Saline was given to the control and model groups. On day 7, the mice were anaesthetized through intraperitoneal injection of sodium pentobarbital (50 mg/kg b.w.) and infected through tracheal intubation with mice from the model group and the administered group (100 μL/each mouse, APP bacterial concentration of 1 × 10^8^ CFU/mL), while the control group was injected with an equal volume of sterile saline. The daily number of deaths, body weight, and feed intake of the animals were recorded. The mice in each group were anesthetized and executed 48 h after successful modelling, and different samples were taken using the following steps.

### 4.4. Bronchoalveolar Lavage Fluid (BALF) Preparation and Analysis

Three mice were randomly picked from each group and fixed on the dissection table in the supine position. The skin, subcutaneous tissue, and muscular layer were cleaned and sequentially incised about 5 mm above the sternal pedicle, the lungs were exposed, and the right lobe of the lung was ligated. The trachea was carefully cut into with an oblique incision. The tracheal tube was introduced and placed 10–15 mm above the tracheal eminence and then tied with a fine thread. The lungs were rinsed with pre-cooled PBS (refrigerated at 4 °C in advance) and rinsed three times, 0.5 mL each time, for about 1.2 mL of irrigating fluid (80% recovery rate). The BALF was then stained with Swiss-Giemsa reagent solution, and the total cell and neutrophil counts were calculated using a blood cell counter. Then, the BALF was centrifuged at 4 °C, 5000× *g* for 10 min, and the supernatant was taken to detect the concentration of C-reactive protein (CRP).

### 4.5. Histopathologic Examination

Fresh mouse lung tissues were fixed in 4% paraformaldehyde for 3 d, dehydrated, and embedded in paraffin wax. The wax blocks were cut into 5 μm thick sections with a microtome, stained with haematoxylin–eosin (H&E), and finally sealed with neutral gum. The sections were placed under a microscope (DM 4000B, Leica, Wetzlar, Germany) to observe the mouse lung tissue. The extent of lung injury and inflammatory cell infiltration was analysed using ImageJ software.

### 4.6. Quantitative Real-Time PCR Analysis

Total RNA was isolated from cells or lung tissues using TRIzol reagent (15596026, Invitrogen Life Technologies, Carlsbad, CA, USA), RNA was reverse transcribed into cDNA according to the PrimeScript RT Reverse Transcription Kit (RR047Q, Takara, Dalian, China), and mRNA expression was quantified using the Applied Biosystems Real-Time Fluorescence Quantitative PCR System and the SYBR premix Ex Taq II (R075A, Takara, Dalian, China). Quantification results were calculated using the 2^®−ΔΔCt^ method compared to β-actin as the reference mRNA (Table 1).

### 4.7. Western Blotting Analysis

Collected lung tissues or cells were lysed with lysate supplemented with PMSF, and then centrifuged at 4 °C for 10 min at 15,000× *g*. The supernatant was collected as protein extracts, and the concentration of protein samples was measured according to the bicinchoninic acid assay (BCA). Equal protein samples were separated using 8–12% SDS-PAGE and then electrotransferred into a nitrocellulose membrane (Millipore, Billerica, MA, USA). The membranes were immersed in 5% skimmed milk powder and enclosed at room temperature for 1 h. The membranes were incubated with NLRP3 (1:1000), Caspase-1 (1:1000), pro-Caspase-1 (1:1000), ASC (1:1000), NF-κB P65 (1:1000), NF-κB p-P65 (1:1000), FL- GSDMD (275) (1:1000), GSDMD (1:1000), and IL-18 (1:1000). Then HRP-coupled goat anti-mouse IgG secondary antibody (1:5000) or HRP-coupled goat anti-rabbit IgG secondary antibody (1:5000) was added and incubated at 37 °C for 1 h. The luminescent solution was configured by mixing WesternBright Sirius HRP substrate’s solution A and solution B in a ratio of 1:1. The moisture on the surface of the membrane was sucked off with filter paper, and the luminescent solution was added dropwise. Ultimately, the membranes were visualized with the chemiluminescence detection kit and examined with a Biosciences Imager. GAPDH or Lamin-B served as the internal control normalized against the overall proteins. Further, the phosphorylated proteins were normalized against their respective total proteins.

### 4.8. ELISA Test

(1) The collected lung tissues from different groups of mice were removed, about 1 g was placed in 800 μL of pre-cooled TRIzol reagent, and an appropriate amount of zirconium oxide beads were added to homogenize the tissue in a tissue homogenizer. After centrifugation for 10 min at 4 °C, 3000 r/min, the supernatant was collected. The levels of TNF-α, IL-1β, IL-18, IL-6, and MPO were determined by referring to the instructions of the mouse ELISA kit.

(2) Cytokine assay: The collected cell supernatant was kept on ice and immediately added to the enzyme wells according to the instructions, and the absorbance value was read at 450 nm on the enzyme labeller (Hong Kong Genetics Limited, Hong Kong, China). The detailed steps were the same as above.

### 4.9. Immunofluorescence Staining

PAMs were placed on coverslips in 24-well plates and left to fuse to 70% for subsequent experiments. After treatment, the cells were treated with NAR for 1 h before APP infection and collected 30 min after APP infection for subsequent experiments. To the cell culture plate, 1 mL of 4% paraformaldehyde was added for fixation, 0.03% Triton X-100 was permeabilised for 20 min at room temperature, and 5% BSA was bound for 30 min at room temperature, after which the binding solution was discarded. The cells were then incubated overnight with P65 primary antibody (1:100) at 4 °C. After washing three times with PBS, the cells were incubated with Alexa Fluor 488 coupled to goat anti-mouse secondary antibody (1:1000) at room temperature for 1 h. Before the end of the incubation, 1 μg/mL of blue fluorescent DAPI dye was added to the plate and incubated for 10 min to stain the nuclear DNA of the cells. Using an IF laser scanning confocal microscope (German Zeiss Z800), the cells were observed at 488 nm (green) and 405 nm (blue) excitation wavelengths for cell fluorescence images.

### 4.10. Co-Immunoprecipitation

Cells were lysed with RIPA lysis buffer, and 50% proteinA/G-agarose working solution was prepared using PBS; 50% proteinA/G-agarose working solution was added to the samples at the rate of 100 μL proteinA agarose beads per 1 mL, and the samples were shaken for 10 min at 4 °C on a horizontal shaker to remove the non-specifically bound proteins. Specifically bound proteins were centrifuged at 4 °C, 14,000× *g* for 15 min, and the supernatant was transferred to a new centrifuge tube to remove proteinA/G-agarose microspheres; after measuring the concentration of the protein samples using a BCA, a particular volume of primary antibody was added. The antigen–antibody mixture was shaken slowly with a shaker, and the model was incubated at 4 °C overnight; the precipitate was collected through centrifugation at 14,000× *g* rpm and washed three times with pre-cooled PBS; the supernatant was collected, and protein denaturation was carried out by adding PAS and Protein Sample Loading Buffer (Denaturing, Reducing, 5×) in a metal bath at 100 °C for 10 min. The supernatant was collected and subjected to SDS-PAGE.

### 4.11. Statistical Analysis

The results were analysed using various statistical tests in GraphPad Prism version 9 (GraphPad Software, San Diego, CA, USA). All the data were statistically analysed through one-way ANOVA using IBM SPSS 27 (SPSS Inc., Chicago, IL, USA). Values were expressed as mean ± SEM. Differences between groups in body-related parameters were analysed using a one-way analysis of variance (ANOVA) and Tukey’s post hoc test. For all tests, a probability value (*p*) of less than 0.05 (*p* < 0.05) was considered statistically significant. Different lowercase letters on the error bars indicate statistically significant differences (*p* < 0.05).

## 5. Conclusions

In short, this study has revealed that NAR has the potential to alleviate the inflammatory response caused by APP by inhibiting the activation mechanism of the NLRP3 inflammasome. It also provides a theoretical basis for the use of NAR as an animal feed additive in the future. Thus, we can fully use the “vaccine-antibiotic-natural drug” triangular defence system to prevent and control lung bacterial infections in pig farms and promote healthy breeding and a safe food supply.

## Figures and Tables

**Figure 1 ijms-25-01027-f001:**
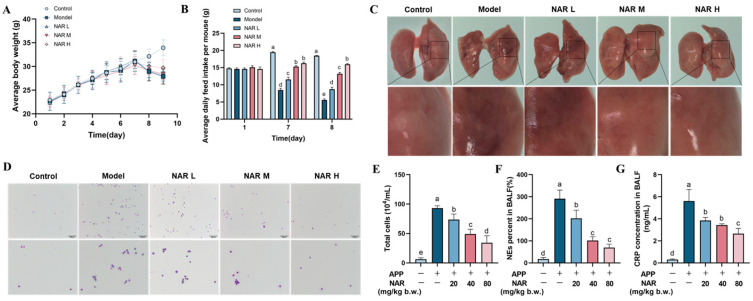
Effects of NAR on APP-induced lesions in mice. (**A**) Effect of NAR on weight loss in APP-induced mice. (**B**) Effect of NAR on food intake of mice. (**C**) Effect of NAR observed during lung dissection after APP infection. (**D**) Observation of Swiss-Giemsa staining after BALF smears of mice (100 μm). (**E**) Effect of NAR on the total number of cells in APP-induced BALF in mice. (**F**) Effect of NAR on the percentage of NEs in APP-induced BALF in mice. (**G**) Effect of NAR on CRP in BALF of APP-induced mice. (Different lowercase letters indicate statistically significant differences between groups *p* < 0.05, while the same lowercase letters indicate no significant differences between groups *p* > 0.05).

**Figure 2 ijms-25-01027-f002:**
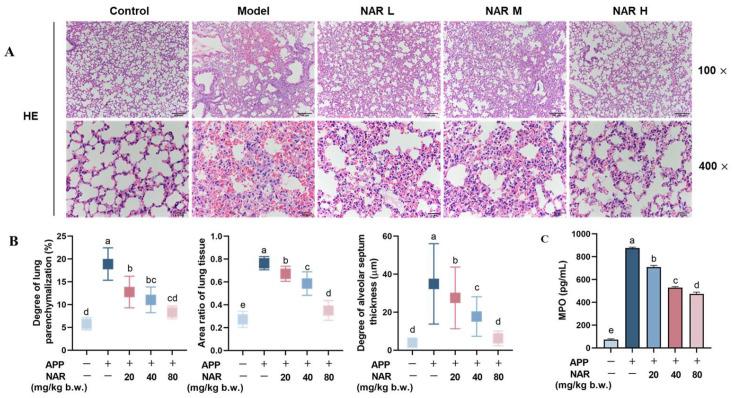
Effect of NAR on APP-induced histopathological damage of lung tissue. (**A**) Pathological tissue observation of mouse lung tissue under H&E staining. (**B**) Effect of NAR on the degree of inflammatory cell infiltration, lung parenchyma, and alveolar wall thickness of mouse lung tissue. (**C**) Effect of NAR on the content of MPO in lung tissue detected by ELISA. Scale bar: 100 μm (100×), 20 μm (400×). (Different lowercase letters indicate statistically significant differences between groups *p* < 0.05, while the same lowercase letters indicate no significant differences between groups *p* > 0.05).

**Figure 3 ijms-25-01027-f003:**
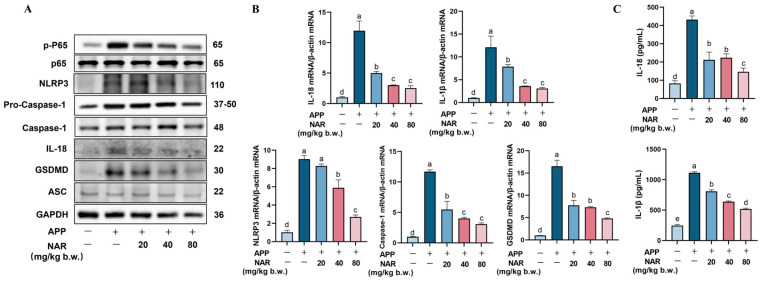
Effect of NAR on APP-induced NLRP3 inflammasome activation in mice lung tissue. (**A**) Effect of NAR on the expression of essential proteins in the NLRP3 inflammasome signalling pathway in lung tissue after APP infection. (**B**) Effect of NAR on the expression of critical genes in the NLRP3 inflammasome signalling pathway in lungs after APP infection. (**C**) Effect of NAR on the expression of IL-18 and IL-1β proteins in lung tissues, shown by ELISA assay. (Different lowercase letters indicate statistically significant differences between groups *p* < 0.05, while the same lowercase letters indicate no significant differences between groups *p* > 0.05).

**Figure 4 ijms-25-01027-f004:**
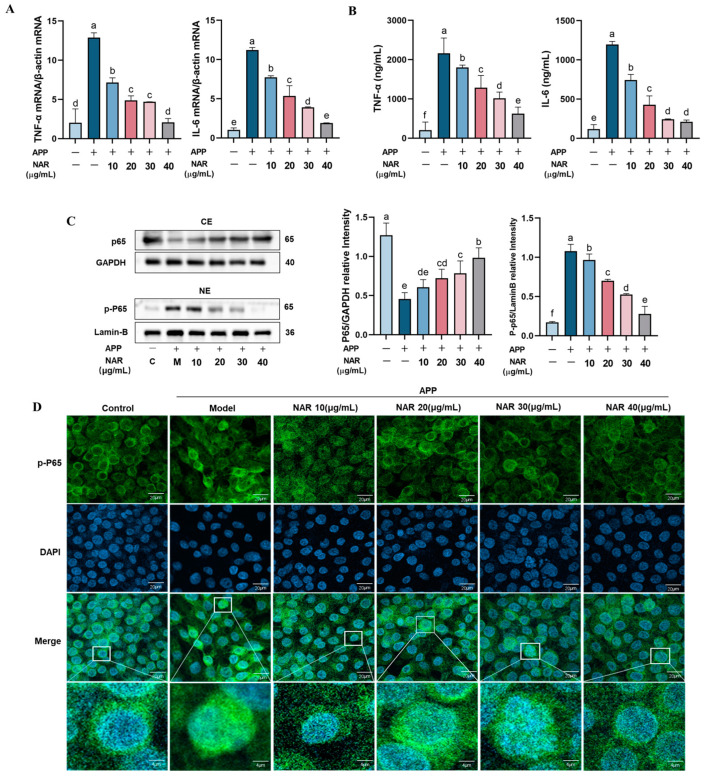
Effect of NAR on APP-induced nuclear translocation of P65 protein in PAMs. (**A**) Effect of NAR on relative mRNA expression of TNF-α and IL-6 in APP infection-induced PAM cells. (**B**) ELISA detection of TNF-α and IL-6 in PAM cells. (**C**) Effect of NAR intervention on APP-induced P65 protein nuclear translocation. (**D**) Confocal microscopy shows that NAR can reduce the nuclear load of P65 in APP-induced PAM cells. Note: Green fluorescence represents P65, blue fluorescence represents DAPI. Scale bar indicates 4 and 20 μm. (Different lowercase letters indicate statistically significant differences between groups *p* < 0.05, while the same lowercase letters indicate no significant differences between groups *p* > 0.05).

**Figure 5 ijms-25-01027-f005:**
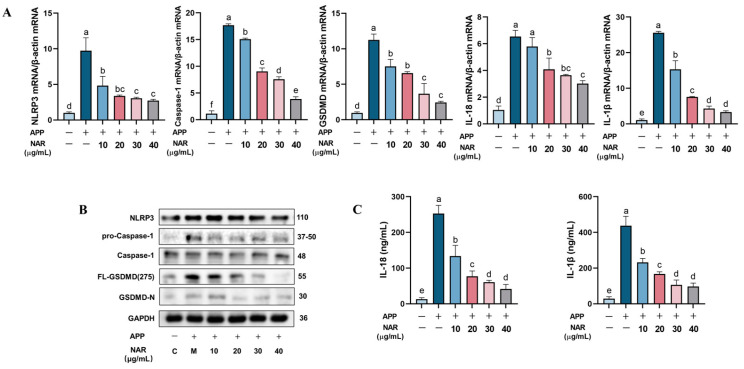
Effect of NAR on the APP-induced NLRP3 inflammasome signalling pathway in PAMs. (**A**) Effect of NAR on the expression of critical genes of the NLRP3 inflammatory vesicle signalling pathway in PAMs. (**B**) Effect of NAR on the expression of essential proteins of the NLRP3 inflammasome in PAMs. (**C**) Effect of NAR on the expression of IL-18 and IL-1β proteins in PAMs. (Different lowercase letters indicate statistically significant differences between groups *p* < 0.05, while the same lowercase letters indicate no significant differences between groups *p* > 0.05).

**Figure 6 ijms-25-01027-f006:**
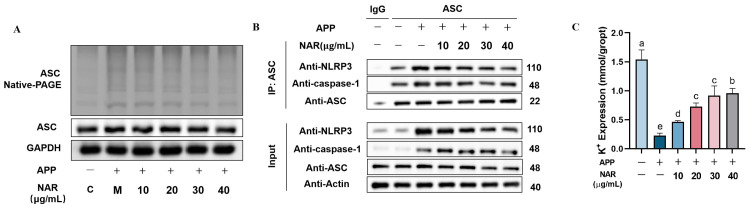
Effect of NAR on assembly of NLRP3 inflammasome complex proteins in PAMs induced by APP. (**A**) Effect of NAR on the level of ASC oligomerization in APP-induced PAMs. (**B**) Effect of NAR on the assembly of complex proteins of the NLRP3 inflammasome in APP-induced PAM cells. (**C**) Effect of NAR on intracellular K^+^ efflux. (Different lowercase letters indicate statistically significant differences between groups *p* < 0.05, while the same lowercase letters indicate no significant differences between groups *p* > 0.05).

**Figure 7 ijms-25-01027-f007:**
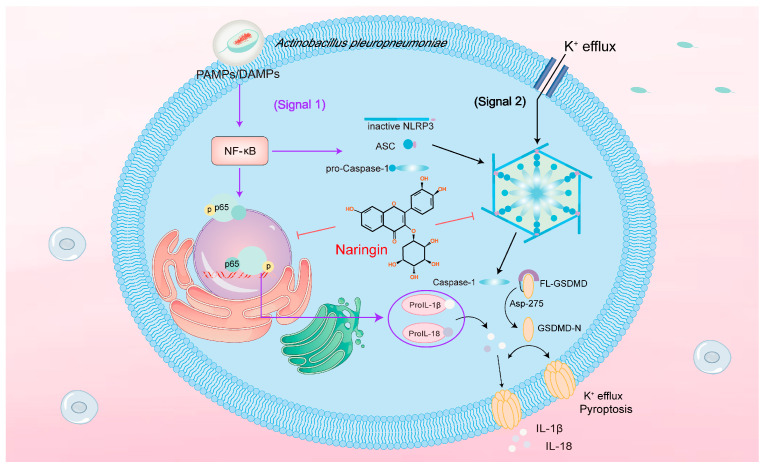
Effect of NAR on APP-induced activation of the NLRP3 inflammasome signalling pathway. The activation of the NLRP3 inflammasome is a dual-signalling activation mechanism. Firstly, APP enters the organism and activates NF-κB via pathogen-associated molecular patterns (PAMPs) or damage-associated molecular patterns (DAMPs). This prompts the entry of the P65 subunit into the nucleus for transcription, thereby promoting the transcription and release of inflammatory factors such as NLRP3, pro-Caspase-1, pro-IL-1β, and other inflammatory cytokines. Transcription and release is the preparatory stage of NLRP3 inflammatory vesicle activation. The second signal is stimulated by APP, K^+^ efflux, or other factors, which activates the assembly of the NLRP3 inflammasome and, in turn, elicits pro-Caspase-1 to form activated Caspase-1. The activated Caspase-1 can cause cellular death by cleaving Gasdermin D and can activate the NLRP3–ASC–Caspase-1 pathway to increase the maturation and release of IL-1β and IL-18. It was found that NAR could alleviate the APP-induced inflammatory response by inhibiting the dual signalling activation mechanism of the NLRP3 inflammasome.

**Table 1 ijms-25-01027-t001:** Primer sequences of the target genes.

Target	Sequence (5′-3′)	Gene Bank Accession No.
IL-1β	5′-GCAGGCAGTATCACTCATTGT-3′ 5′-GGCTTTTTTGTTGTTCATCTC-3′	LT727274.1
IL-6	5′-CTGCAGTCACAGAACGAGTG-3′ 5′-GACGGCATCAATCTCAGGTG-3′	XM_047753916.1
TNF-α	5′-AAGGGAGAGTGGTCAGGTTGC-3′ 5′-CAGAGGTTCAGTGATGTAGCG-3′	NM_001278601.1
IL-10	5′-GCTCCTAGAGCTGCGGACTGC-3′ 5′-TGCTTCTCTGCCTGGGGCATCA-3′	XM_021175612.1
IL-18	5′-GCTTGAATCTAAATTATCAGT-3′ 5′-GAAGATTCAAATTGCATCTTAT-3′	EF444989.1
NLRP3	5′-AGACCTCCAAGACCACTAC-3′ 5′-ACATAGCAGCGAAGAACTC-3′	XR_004607107.1
Caspase-1	5′-TGCCCAGAGCACAAGACTTC-3′ 5′-TCCTTGTTTCTCTCCACGGC-3′	XM_032910338.1
β-actin	5′-GGTCACCAGGGCTGCTTT-3′ 5′-ACTGTGCCGTTGACCTTGC-3′	OY781925.1

## Data Availability

The data presented in this study are available upon request from the corresponding authors.

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
