# Peer review of "Naringin’s Alleviation of the Inflammatory Response Caused by Actinobacillus pleuropneumoniae by Downregulating the NF-κB/NLRP3 Signalling Pathway"

_ijms, 2024, doi:10.3390/ijms25021027_

Round 1
Reviewer 1 Report
Comments and Suggestions for Authors
Reviewer comments
Manuscript ID: ijms-2792951
Title: Naringin alleviated the inflammatory response infected with Actinobacillus pleuropneumoniae by downregulating the NF-κB / NLRP3 signalling pathway
The present study investigated investigated the effects of Naringin (NAR) on the inflammatory response caused by Actinobacillus pleuropneumoniae (APP) through both in vivo and in vitro models. The study concluded that that NAR ameliorates inflammatory injury in the lungs due to APP induction by inhibiting the NF-κB/ NLRP3 signaling pathway.
This study is quite interesting and extends the horizon for NAR as a potential natural product for preventing and treating APP. I have minor points would like the author to discuss.
Reviewer comments
Introduction
· Line 62: “Recent studies have revealed that NAR has many beneficial consequences, including anti-inflammatory, antioxidant, and anti-apoptotic responses [9].’ Please mention more than one study.
Results
· Please revise the significance in section “Effect of NAR on histopathological damage in the lungs of mice after APP infection’” and Figure 2. Revise the significance in all three
the degree of inflammatory cell infiltration, lung parenchymatous and alveolar wall thickness. As from the figure, there is no significant difference between the three does of NAR, still the authors wrote different symbols indicating significant differences.
Material and methods:
· Why did the authors choose Kunming mice as a type of mice in this experiment. Is there a specific reason?
· On what basis did the authors choose the doses of naringin? for instance, why 20 is considered as the low dose?
· The authors mentioned that “ infected by tracheal intubation with mice from the model group and the administered group (100 μL /each mouse, APP bacterial concentration of 1 × 108 CFU/mL). On what basis did the authors choose this amount will induced lung infection.
· Please add accession no for the Primer sequences used of the target genes.
Conclusion
· The conclusion section is written as 5.Results. Please correct it

Comments on the Quality of English LanguageMinor editing of English language required
Reviewer 2 Report
Comments and Suggestions for Authors
The manuscript is good enough. The English is correct. The figures and their presentation are good. The figures can be understand. Only minor revisions will be added.
1) Sometimes the Western blots have extra bands. Can you explain it.
2) The cells in the images are not different between treatments. Can you add better images.
3) If Inflammation is determinated, probably more cytokinesis or chemokynes must be assays.
4) Authors need to mid when will be toxic the addition of the product. Every drug can be toxic at high level.
5) Some words are bad written
Round 2
Reviewer 2 Report
Comments and Suggestions for Authors
Accept in this version